# Structural and Functional Insights into Human Nuclear Cyclophilins

**DOI:** 10.3390/biom8040161

**Published:** 2018-12-04

**Authors:** Caroline Rajiv, Tara L. Davis

**Affiliations:** 1Department of Biochemistry and Molecular Biology, Drexel University College of Medicine, Philadelphia, PA 19102, USA; caroline.rajiv@gmail.com; 2Janssen Pharmaceuticals Inc., 22-21062, 1400 McKean Rd, Spring House, PA 19477, USA; 3FORMA Therapeutics, 550 Arsenal St. Ste. 100, Boston, MA 02472, USA

**Keywords:** peptidyl prolyl isomerases, nuclear cyclophilins, spliceophilins, alternative splicing, spliceosomes, NMR, X-ray crystallography

## Abstract

The peptidyl prolyl isomerases (PPI) of the cyclophilin type are distributed throughout human cells, including eight found solely in the nucleus. Nuclear cyclophilins are involved in complexes that regulate chromatin modification, transcription, and pre-mRNA splicing. This review collects what is known about the eight human nuclear cyclophilins: peptidyl prolyl isomerase H (PPIH), peptidyl prolyl isomerase E (PPIE), peptidyl prolyl isomerase-like 1 (PPIL1), peptidyl prolyl isomerase-like 2 (PPIL2), peptidyl prolyl isomerase-like 3 (PPIL3), peptidyl prolyl isomerase G (PPIG), spliceosome-associated protein CWC27 homolog (CWC27), and peptidyl prolyl isomerase domain and WD repeat-containing protein 1 (PPWD1). Each “spliceophilin” is evaluated in relation to the spliceosomal complex in which it has been studied, and current work studying the biological roles of these cyclophilins in the nucleus are discussed. The eight human splicing complexes available in the Protein Data Bank (PDB) are analyzed from the viewpoint of the human spliceophilins. Future directions in structural and cellular biology, and the importance of developing spliceophilin-specific inhibitors, are considered.

## 1. Introduction

Cyclophilins are members of the peptidyl prolyl isomerase (PPI) family, along with the structurally unrelated FK506 binding proteins and the parvulins (EC 5.2.1) [1,2,3]. Cyclophilins are evolutionarily widespread, with paralog cyclophilins found in all kingdoms of life, including viruses. In many species, there exist multiple cyclophilins encoded in the genome; depending on how strictly one defines a sequence as cyclophilin-like, there are anywhere from 17 to 30 cyclophilins encoded in humans [1,2,4]. In humans, as in other multicellular organisms, when multiple cyclophilin family members are encoded, they are also often found targeted to multiple cellular compartments. One family member, peptidyl prolyl isomerase F (PPIF), is targeted to mitochondria, where it participates in the regulation of the mitochondrial permeability transition pore [5,6,7]. The three cytoplasmic cyclophilins—the canonical family member peptidyl prolyl isomerase A (PPIA), plus peptidyl prolyl isomerase B (PPIB) and peptidyl prolyl isomerase C (PPIC)—have also been detected extracellularly in various reports. These studies have focused on the mechanisms by which these three cyclophilins facilitate host: viral interactions, particularly for human immunodeficiency virus (HIV) and hepatitis A, B, and C viruses [8,9,10]. Much less is known about the cytoplasmic cyclophilins peptidyl prolyl isomerase D (PPID), peptidyl prolyl isomerase-like 4 (PPIL4), RAN binding protein 2 (RANBP2), and the nuclear cyclophilins peptidyl prolyl isomerase-like 4 (PPIL6) and natural killer cell triggering receptor (NKTR), and these will not be discussed further here. The purpose of this review is to highlight what is currently known about the structure and biological function of the eight human nuclear-localized cyclophilins: peptidyl prolyl isomerase E (PPIE), peptidyl prolyl isomerase G (PPIG), peptidyl prolyl isomerase H (PPIH), peptidyl prolyl isomerase-like 1 (PPIL1), peptidyl prolyl isomerase-like 2 (PPIL2), peptidyl prolyl isomerase-like 3 (PPIL3), peptidyl prolyl isomerase domain and WD repeat-containing protein 1 (PPWD1), and spliceosome-associated protein CWC27 homolog (CWC27).

Cyclophilins are characterized by a central closed barrel type beta fold, with large regions of ordered loops and two small helices packed against the loop region and sheets β7–β8 of the central barrel (structural classification of proteins (SCOP) fold family 50892). See Figure 1 for the structure of the canonical family member PPIA. There is a high degree of sequence and structural similarity across the entire cyclophilin family, which has been discussed elsewhere [1,3,4]. A simple sequence alignment comparing the isomerase domains of the eight nuclear cyclophilins to PPIA, along with a structure-based alignment, is shown in Appendix A.

Additionally, many cyclophilin family members in more complex organisms encode for additional motifs and domains (see Figure 2), discussed in further detail below for the human nuclear cyclophilins PPIE, PPIG, PPIL2, PPWD1, and CWC27. Originally characterized as the protein receptor for the natural product cyclosporine, cyclophilins have long been studied in vitro because of their high solubility and attractive biophysical properties. Decades of structural work utilizing well-behaved cyclophilins have led to nearly complete coverage of the human cyclophilin family, by both X-ray crystallography and nuclear magnetic resonance (NMR) [1,11,12,13,14,15]. These structures have been used to facilitate studies focused on substrate specificity and selective pharmacophores, persistent issues in the field due to the high degree of sequence identity within the S1 pocket (Figure 1B and Appendix A). It is likely that specificity in this family, if focused on the active site, will depend on rational design focusing on the less conserved S2 pocket, contiguous to S1 (Figure 1A,C) [8,16]. An opinion on the utility of design against the S2 pocket for the nuclear cyclophilins will be addressed at the end of this review.

When the technology to purify and subject splicing complexes to proteomics study was developed, it quickly became clear that the cyclophilins localized to the nucleus could accurately be called “spliceophilins”. While researchers were describing in ever greater detail the role of splicing subcomplexes in regulating pre-mRNA splicing, the presence of all eight nuclear cyclophilins, and the mystery of what peptidyl prolyl isomerases might be doing in this RNA-driven process, were noted by several groups [17,19,20,21,22]. In Figure 3, a typical splicing cycle is presented; as can be seen, structures now exist for every step of the splicing cycle. These structures have contributed greatly to our knowledge of how RNA and protein work in concert to splice together exons and remove intronic sequence from single-stranded mRNA. A subset of these structures also place spliceophilins squarely in the center of the action, sometimes quite literally.

Messenger RNA is processed by the spliceosome, the complex and dynamic macromolecular machine that removes intronic sequences and joins coding sequence together. This machine is composed of five uridine-rich small nuclear RNAs (snRNAs) which interact with proteins to form snRNP subcomplexes. The RNAs perform the two trans-esterification reactions associated with intron removal and exon ligation and are also critical to substrate recognition and quality control processes (Figure 3). The roles of spliceosomal proteins are less well understood, other than those that serve as chaperones and protectors of spliceosomal RNA [20,21,22,23]. Small nuclear RNAs and associated proteins are largely conserved in all eukaryotes, but the spliceosomes of complex eukaryotic branches contain hundreds of additional protein factors. These proteins, often called “accessory” to the spliceosome, are abundantly found associated with spliceosomes at the early, intermediate, and catalytic stages of splicing [17,18,19]. Accessory proteins are often conserved across species and are likely to play regulatory roles in the organisms that encode for them. However, many accessory proteins do not have orthologs in model genetic organisms, and are missing from the budding yeast *Saccharomyces cerevisiae*. This lack of conservation limits the use of simple genetic organisms to study the function of accessory proteins and complicates assay development to screen for small molecule inhibitors that might modulate alternative splicing.

Spliceophilins are generally considered to be accessory to the spliceosome, although considering their presence throughout the splice cycle, this term may downplay their role in regulation of splicing. To date it is structural information for complexes containing the nuclear cyclophilins, rather than structures of the cyclophilins alone, that have been most informative in studying the regulation of transcription and of pre-mRNA splicing [1,15,16,30,31,32,33,34,35]. There are multiple cryo-electron microscopy (cryo-EM) structures of spliceosomal complexes that contain one or more of the nuclear cyclophilins; with some technical caveats, these structures can also add crucial information to the ways in which this subset of the cyclophilin family may regulate nuclear processes (Figure 3) [24,25,26,27,28,29]. A recent review covers the overall insights into pre-mRNA splicing gleaned from spliceosomal structure; this review will focus exclusively on the spliceophilins and their unique interactions within the spliceosome [36]. It will hopefully become clear to the reader that cyclophilins participate uniquely in nuclear function, despite their apparent structural similarity. We will present each cyclophilin individually to allow us to highlight their unique nature.

## 2. Background and Structures of the Nuclear Cyclophilins within Splicing Complexes

The domain organization of the nuclear cyclophilins, along with the canonical family member PPIA, and their association with splicing complexes and subcomplexes, is depicted in Figure 2. Details including accession numbers, domain boundaries, and structural information are collated in Table 1 for the nuclear cyclophilins. Additional information, including full amino acid sequences, can be found in Appendix A. Appendix A contains more detailed information concerning the subset of spliceosomal structures, and the interactions between the cyclophilins in each structure, in greater detail.

### 2.1. Peptidyl Prolyl Isomerase Isoform H 

PPIH is a minimal cyclophilin, meaning that it encodes for a single isomerase domain. In early publications, Cyp-H, USA-Cyp, or U4/U6-20K were used to designate PPIH [38,39]. The apo structure of PPIH was initially published in 2000 and was then solved in complex with a 30-mer peptide derived from its spliceosomal binding partner, pre-mRNA processing factor 4 (PRPF4) (Figure 4A). Note that PRPF4 should not be confused with Prp4, a nuclear kinase whose official name is PRPF4B [40,41,42]. These proteins are unrelated and are often confused with each other. The first cyclophilin that associates with the spliceosome is PPIH; it is present within the stand-alone tri-snRNP complex, and integrates into the splicing machinery as part of the transition into the early intermediate B complex [17,18]. The PPIH protein is short-lived within splicing complexes and is not detectable within the spliceosome during later pre-catalytic B_act_ or B* stages [17]. A structure of complex B containing the first visualization of PPIH in a splicing complex, albeit with the aid of cross-linking reagents, has been recently published [24].

The initial interaction between PPIH and PRPF4 was identified as a result of yeast two-hybrid studies [38,39,41]. It was notable at the time that this interaction occurred distal to the active site of PPIH, did not involve interaction with a target proline in PRPF4, and was unaffected by cyclosporine (Figure 4A). More recently, our group has delved more deeply into the relationship between PPIH and PRPF4 and have discovered that there is a second site of interaction that does involve the active site of PPIH and the N-terminus of PRPF4 [32]. However, rather than being crucial for a conformational change within PRPF4, we propose a model in which this interaction instead is mutually beneficial in protecting PRPF4 from unregulated post-translational modifications in its intrinsically disordered N-terminal region, while also blocking the active site of PPIH from unregulated proline binding and turnover (Figure 4B). This may well be a general phenomenon among the spliceosome-associated cyclophilins, considering the observation of other cyclophilin-centric interactions within the spliceosome that clearly do not involve proline binding outlined later within this review. Unfortunately for those interested in PPIH interactions within the spliceosome, both purified tri-snRNP and B complexes from *Schizomyces pombe* and from human tissue culture lines have been shown to contain PPIH in solution, but fail to visualize the protein in the resulting structures. However, a recent structure of the precatalytic B complex captured PPIH via crosslinking, and the authors were able to model the interaction between the critical phenylalanine residue from PRPF4 and the α1–β3 loop of PPIH, thereby recapitulating what had previously been seen in the binary structure (Figure 4C) [24]. The N-terminal region of PRPF4 was not modeled, and so the second site characterized in Rajiv et al. [32] could not be confirmed (Figure 4). However, PPIH is near three additional spliceosomal proteins not yet studied for PPIH interaction in vitro: WW domain binding protein 4 (WBP4), pre-mRNA processing factor 6 (PRPF6), and pre-mRNA processing factor 8 (PRPF8). These interactions might prove critical to understanding the function of PPIH in regulation of splicing, and we await further characterization of these interactions with purified proteins in vitro.

### 2.2. Peptidyl Prolyl Isomerase Isoform E

PPIE is a multidomain cyclophilin, encoding for an N-terminal RNA recognition motif (RRM) and a C-terminal isomerase domain (Figure 5A). Initial studies referred to PPIE as Cyp33. The structure of the isomerase domain of PPIE at 1.61 Å was first solved and deposited as part of a structural genomics initiative, the Structural Genomics Consortium (SGC) [43], later described in Davis et al. [1]. A slightly lower-resolution structure (1.88 Å) was independently deposited and described in a publication by Wang et al. [44], and a lower-resolution structure (2.5 Å) of the same domain was deposited once again in 2011 by the Joint Center for Structural Genomics (JCSG) [45]. Likewise, a structure of the RRM motif of PPIE was originally solved as part of the RIKEN initiative [46], and five years later independently published by three groups [47,48,49]. Finally, a chimeric structure of the RRM motif of PPIE fused to the plant homeodomain (PHD) motif of mixed-lineage myeloma-1 (MLL1) has been published (Figure 5B) [47]. All PPIE structures, whether of the isomerase domain or of the RRM motif, are largely superimposable (Figure 5A). There are several cyclophilins that participate in nuclear processes independent of mRNA splicing, including PPIE. A series of studies in 2010 found that both the RRM and isomerase domain of PPIE participated in chromatin remodeling complexes [44,47,48,49]. Using in vitro assays, structural analysis, and cellular assays, the RRM motif was found to interact directly with the PHD3 domain in the histone reader MLL1, while the isomerase domain interacts with a proline in MLL1 to allow for the RRM–PHD complex to form. The RRM interaction surface was found to be extensive, including residues from all four central β-sheets in the RRM (Figure 5B) [47,48,49]. A separate study identified PPIE as part of the XAB2 complex, composed of a subset of spliceosomal proteins: XPA binding protein 2 (XAB2)/SYF1, aquarius (AQR), zinc finger protein 830/coiled-coil domain containing isoform 16 (ZNF830/CCDC16), ISY1, and PPIE that binds RNA but also participates in transcription-coupled DNA repair in cells [50]. The XAB2 complex was later termed the intron binding complex (IBC), and the RNA interaction was characterized in much greater detail [51,52]. No specific interactions between AQR and PPIE, or other proteins within the IBC, have been validated in vitro.

The nineteen complex (NTC), also called PRP19 complex, associates with the spliceosome as part of the conversion to the pre-catalytic B complex [17,22]. As with other PRPF19 complex members, PPIE is found in B complex, is highly abundant in the later intermediate complexes B* and B_act_, and is also abundant in the catalytic C complex [17,18]. Versions of all three of these complexes have been published from *S. pombe* and human, but PPIE has only been visualized in human complexes [26,27,28].

In human spliceosomes, PPIE has been built into four structures: two representing pre-catalytic B complexes, and two representing the catalytic C complex [26,27,28]. Both the RRM and isomerase domains are modeled into density in all these structures, providing novel insight into the particular interactions mediated by each domain. In the mature B_act_ structure, the region around Pro83 of splicing factor 3B, subunit 4 (SF3B4) is pointed towards the active site of PPIE (Figure 5C). Other interactors include splicing factor 3B, subunit 2 (SF3B2), also with the catalytic face of PPIE, and splicing factor 3A, subunit 2 (SF3A2), which interacts both with the cyclophilin and RRM domain of PPIE. In addition to extensive contact with SF3A2, the RRM motif is also potentially interacting with SYF1. These interactions are largely with the α-helix and loop regions of RRM, as opposed to the β-sheet mediated interaction with MLL1 (Figure 5B). In the late B_act_ spliceosome, this protein network surrounding PPIE is largely unchanged.

Finally, the two catalytic complexes that model PPIE do so in very different orientations. In the step 1 catalytic C complex, the isomerase domain of PPIE is interacting with ISY1, small nuclear ribonucleoprotein-associated protein A’ (snRNPA’), and small nuclear ribonucleoprotein-associated proteins B and B’ (snRNPB/B’), with small nuclear ribonucleoprotein-associated protein D3 (SmD3) also closely associated with snRNPA’ and B/B’ (Figure 5D,E). The RRM interaction with SYF1 is preserved. Visible in this structure is the position of the pre-mRNA substrate, which seems to interact directly with the long β2–β3 loop in the RRM (Figure 5E). Also interacting with the RRM motif is AQR and snRNPA’. In the C* complex spliceosome published in Bertram et al. [27] the interaction with SYF1 and AQR is mediated through the isomerase domain of PPIE rather than the RRM motif, and the model of U2 small nuclear ribonucleic acid (U2 snRNA) is placed extremely close to PPIE (Figure 5F). In this structure the orientation of PPIE seems to be flipped, with the isomerase domain interacting with AQR and SYF1, and the RRM motif close to pre-mRNA processing factor 17 (PRPF17). Perhaps future structures will resolve this seeming discrepancy.

### 2.3. Peptidyl Prolyl Isomerase-Like Isoform 1

PPIL1 is a minimal cyclophilin, encoding for a single isomerase domain. The structure of PPIL1 was solved via NMR and initially described in [35]. Both Xu et al. [35] and a later study of PPIL1 were focused upon its interaction with the spliceosomal binding partner SNW domain-containing protein 1 (SNW1); also known as SKI-interacting protein (SKIP). In these studies and others, the interaction between PPIL1 and SNW1 was found to involve residues from the β2–α1, β4–β5, and β7–α2 loop regions outside of the S1 proline-binding pocket [15,34,35]. All these structures overlay nicely, considering they are derived from both crystallography and NMR (Figure 6A). To date, PPIL1 has has not been isolated as part of any other nuclear complex, and so is assumed to regulate pre-mRNA splicing.

As with PPIE, PPIL1 is an early spliceophilin; it is part of the PRP19 complex, along with SNW1. It is highly abundant in complex B* and complex B_act_ and is also abundant in the catalytic C complex [17,18]. Versions of complexes have been published out of *S. pombe* and human, but PPIE has only been visualized in human structures. Six deposited spliceosomal structures contain PPIL1: the four outlined above that also contain PPIE, as well as an additional representative of B_act_ and C* complexes [24,25,26,27,28,29]. In the mature B_act_ complex, we again see interaction between the S1 pocket and proline, this time with Pro95 of PRPF17 (Figure 6B). All other interactions with PPIL1 are outside of the active site and include not only SNW1, but also RBM22 and SPF27 (Figure 6C). Both cell division cycle 5-like protein (CDC5L) and crooked neck-like protein 1 (CRNKL1) are in the region, but not directly interacting with PPIL1. One novel finding within this spliceosomal structure is the extensive ordering of the disordered region of SNW1 around the β4–β5 and β7–α2 loop regions of PPIL1. All studies with isolated SNW1 polypeptides have been disordered, and so this is the first direct visualization of this interaction. In the B_act_ complex from Haselbach et al. [25] many proteins seen in the mature B_act_ complex are missing from the model, and SNW1 is largely disordered (Figure 6D). However, the S1 interaction with PRPF17, and some of the previously described interactions with SNW1 and RBM22, are preserved [25]. Finally, all three structures representing C complex are very similar, with PPIL1 interactions again with U5 snRNP 40 kDa, PRPF17, CDC5L, RBM22, CRNKL1, and SNW1. However, SYF2 is now interacting with the back-face of PPIL1 near α2 (Figure 6E).

### 2.4. Peptidyl Prolyl Isomerase-Like Isoform 2

PPIL2 is a multidomain cyclophilin, encoding for an N-terminal U-box motif and a C-terminal isomerase domain (Figure 2). Aliases for PPIL2 include Cyp-60, RING-Type E3 ubiquitin transferase isomerase-like 2, or UBOX7. The structure of the isomerase domain of PPIL2 was solved at the SGC and published (Figure 7A) [1]. There is no structure of the PPIL2 U-box publicly available, although it is very similar to other U-box motifs of the C_2_H_2_ type, including the spliceosomal protein PRPF19 (Figure 7A). The U-box of PPIL2 is active in vitro, and has been reproducibly shown to perform as a functional E3 ligase when coupled to ubiquitin, the canonical E1 UBA1, and a nuclear E2 (Figure 7B) [53,54]. No full-length structure of PPIL2 is available, and the physiological relevance of PPIL2 E3 ligase activity remains unclear—although it is notable that splicing activity and the assembly of spliceosomal complexes are likely regulated by ubiquitination [37,55,56,57,58,59]. Two spliceophilins (PPIL2 and CWC27) have a naturally occurring substitution in the S1 pocket that renders them incapable of prolyl isomerization, although still capable of binding proline. In PPIL2, the canonical S1 residue Trp121 is replaced by a Tyr, resulting in a loss of both isomerase activity and affinity for the pan-inhibitor cyclosporine [1]. There is a unique set of four proteins that associate transiently with the spliceosome during the transition from B_act_ to catalytic C complex, including PPIL2 [17,18]. Consequently, PPIL2 is highly abundant only in B_act_. Unsurprisingly, due to the highly transient nature of this subcomplex of the spliceosome, PPIL2 is not modeled in the existing structures of pre-catalytic spliceosomes.

There is both pull-down and yeast two-hybrid data indicating that PPIL2 interacts with the spliceosomal proteins zinc finger 830 (ZNF830) and proline-rich protein PRCC (PRCC) (UniProtKB Entry Q96NB3 and Q92733, respectively) [60]. We have prepared soluble forms of full-length versions of both PPIL2 and ZNF830 proteins (Figure 7B). We have also isolated soluble forms of the two domains of ZNF830, along with the two domains of PPIL2. We then validated that full-length PPIL2 directly interacts with reasonable affinity (≈500 nM) to full-length ZNF830 (Figure 7C). Interestingly, both the isolated U-box motif and the isomerase domain of PPIL2 can mediate the interaction with ZNF830 [61]. Additionally, we have evidence that both domains of PPIL2 simultaneously interact with ZNF830 (Figure 7D). Further work to delineate the functional significance of these interactions is ongoing, but certainly this mode of binding is unique among what we have seen for other spliceophilins.

### 2.5. Peptidyl Prolyl Isomerase-Like Isoform 3

PPIL3 is a minimal cyclophilin. Initial studies called PPIL3 CypJ [62]. The structure of the isomerase domain of PPIL3 was solved via crystallography and initially described in [52]. A later structure of PPIL3 in apo form, and bound to cyclosporine, is also available in the PDB (Figure 8). Several spliceophilins, including PPIL3, PPIL2 and CWC27, are first found in B_act_ complex. However, unlike PPIL2, PPIL3 is most abundant in C complex [17,18]. Although multiple cryo-EM structures of B_act_ and C complex exist, PPIL3 has not been successfully modeled into density.

### 2.6. CWC27

Also referred to as NY-CO-10, SDCCAG10, or SDCCAG-10, CWC27 is a complex cyclophilin, encoding for an N-terminal isomerase domain and a large C-terminal repetitive low complexity region of unknown function (Figure 2). The structure of the isomerase domain of CWC27 was solved at the SGC, and was initially described in [1]. A later structural study of CWC27, comparing the human CWC27 to that of a thermophilic organism, can be found in [63]. Like PPIL2, CWC27 is naturally substituted in the S1 pocket, with a glutamic acid substitution for tryptophan (Figure 9A). Additionally, CWC27, like PPIL2 and PPIL3, is highly abundant in B_act_ complex, and is also found with moderate abundance in C complex [17,18].

The isomerase domain of CWC27 has been modeled into three spliceosomal complexes [25,26]. In the mature B_act_ complex, U5 snRNP 200 kDa interacts with the α2–β8 region of CWC27; BUD31 is found near the β4–β5 loop; PRPF8 interacts with both the β4–β5 loop and β7–α2 region; and RNF113 (RING finger protein 113) interacts with the C-terminal linker region between the PPI and the low complexity region of CWC27, which is apparently disordered (Figure 9B). In the late B_act_ complex the CWC27 interaction environment is quite similar, save for the absence of BUD31 and slight movement of PRPF8 (Figure 9C). Finally, in the B_act_ complex of Haselbach et al. [25] the modeled interactions are comparable. In this structure, more of RNF113 can be modeled in, including an additional, extensive interaction with β1–β2 of CWC27. The model includes BUD31 but not U5 snRNP 200 kDa is not (Figure 9D) [25]. Note that in all structures, only the isomerase domain of CWC27 can be modeled, leaving roughly 200 residues uncharacterized.

### 2.7. WD40-Domain Containing Peptidyl Prolyl Isomerase 1

PPWD1 is a multidomain cyclophilin, encoding for an N-terminal seven-bladed WD40-domain and a C-terminal isomerase domain (Figure 2). An early alias of PPWD1, still found in some databases, is KIAA0073. The structure of the isomerase domain of PPWD1 was initially solved by the SGC, described in [1,30] (Figure 10A). Interestingly, the crystal structure of the PPWD1 cyclophilin captured an interaction between molecules in the asymmetric unit that was proline-mediated; however, in vitro this isolated peptide was found to interact with PPWD1, but not to be a substrate for turnover (Figure 10B) [30]. We note that it is common in cyclophilin structures to find proline-containing peptides from symmetry-related cyclophilins interacting in the S1 pocket, and is perhaps one of the keys to both the crystallizability of these proteins, and their potential for higher-order aggregation in NMR studies. PPWD1 is highly abundant only in C complex [17,18]. One of the C complex structures contains a model of PPWD1 [28]. In this complex, only PRPF8 is close enough to form contacts with the isomerase domain of PPWD1, centered on the β7–α2 region (Figure 10C). Over 90 Å away, PRPF8 along with U5 snRNP 200 kDa form the neighborhood around the WD40 domain of PPWD1 (Figure 10D). This represents the first, and only, experimental model for this domain.

### 2.8. Peptidyl Prolyl Isomerase Isoform G

PPIG is a multidomain cyclophilin. Much like NKTR, PPIG encodes for an N-terminal isomerase domain and a C-terminal SR-repeat region (Figure 2). Alias of PPIG include SR-Cyp or Clk-associating RS-cyclophilin (CARS-Cyp) [60]. The structure of the isomerase domain of PPIG was solved at the SGC and was published [1]. Later structures of the apo isomerase domain and bound to the pan-inhibitor cyclosporine were also published [64]. These structures are practically indistinguishable (Figure 11A). PPIG is not highly abundant in the spliceosome, but a small number of peptides were identified in C complex [17,18]. In spite of this, PPIG has been successfully modeled into one C complex structure [28]. In this structure, only the isomerase domain of PPIG is modeled, and it interacts with SNW1, modeled as an alpha helix and interacting in the α2–β8 loop region. PPIG is also seen to interact with CWC15 (β3–β4 loop), and PRPF8 (3_10_ helix) (Figure 11B). No model exists for the residues outside the PPIG isomerase domain, meaning roughly 600 residues of PPIG are unaccounted for in the structure.

## 3. Structural Analysis of the Spliceophilins

Due to their soluble nature and ease of crystallization, the spliceophilins have been the targets of extensive biochemical and biophysical characterization. It is somewhat surprising how long it has taken to begin to visualize the protein and RNA neighbors of these proteins in the context of the spliceosome. Now that spliceosomal models exist that contain spliceophilins, what can we learn from these studies? First, it is useful to compare the numerous, publicly available results of proteomic studies to the protein neighbors of the spliceophilins depicted in Figure 4, Figure 5, Figure 6, Figure 7, Figure 8, Figure 9, Figure 10 and Figure 11. The interaction between PPIH and PRPF4 was well-characterized and is preserved within the spliceosome (Figure 4) [32]. On the other hand, PPIH is also found to interact with a whole host of other proteins, both spliceosomal and nuclear, along with cytosolic proteins, in major publicly available databases (BioGRID, IntAct, and STRING, all accessible through UniprotKB) [60]. In addition to PRPF4, which is represented in all three databases, there are interactions reported with other tri-snRNP components (PRPF3, PRPF8, PRPF18, PRPF31); with other spliceosome-associated proteins (hnRNPD, LSM4, LSM6, LSM8, BAG2); and with many other proteins that are seemingly unrelated to splicing complexes (ubiquitin-specific proteases, finbronectin-1, E3 ligases including HUWE1, etc.) [60]. How do we interpret these results, in the context of both extensive biochemical and structural work done on PPIH outside of the spliceosomal context, and also in the context of spliceosome structure? Firstly, many subcomplexes of the spliceosome, including the tri-snRNP, the PRP19 complex, and the IBC, are stable outside the context of the large spliceosome complex and may be isolated using pull-down or two-hybrid approaches [23,65,66]. Therefore, we must always keep in mind that the proteomics data in public databases are sensing not only single protein interactors but are detecting larger subcomplexes as well. Additionally, cyclophilins are very likely to provide both false positive and false negatives in proteomics studies, unless experimental conditions are optimized. This is because, on one hand, proline-mediated interactions and even native substrates of the cyclophilins tend to bind in the low-micromolar range of affinities. These are likely to go undetected in most proteomics screens, as evidenced by the PRPF4 interaction delineated in Rajiv et al. [32] or the lack of any prior data indicating both PRPF17 and SF3A2 as a proline-mediated interactor of spliceophilins. On the other hand, interactions mediated by the α1–β3, β4–β5, and α2 region of spliceophilins seem, based on the limited amount of work reported, to provide much higher affinity binding sites for splicing complex proteins [32,34]. Generally, it seems like the cyclophilin fold can provide a productive set of surfaces centered on the extensive loop structures surrounding the beta-sandwich core, and these interactions may well be promiscuous. This is likely the reason why cytoplasmic proteins are seen to interact with nuclear spliceophilins, when they are unlikely to ever see these proteins in the cellular milieu. Even though several of these types of interactions may now been seen for the spliceophilins in splicing complexes, it is by no means sure that these interactions occur outside the scaffold of the full spliceosome, and it is also unlikely that the current spliceosomal structures contain all the potential interactions between spliceophilins and nuclear proteins that are possible. Without this full list of interactions, a consensus binding sequence to loop regions has to date been elusive, and not enough of these interactions have been biochemically characterized outside of the spliceosomal complex to allow for any predictive power to be applied as a filter to proteomics lists. Finally, a large degree of the diversity within spliceophilins is contained within modules outside of the isomerase domain. However, the relative challenges of expressing and purifying some of these domains (intrinsically disordered SR motifs and regions of low homology to other proteins, WD40 domains) and the lack of domain-specific proteomic studies again slows progress along these lines. When they have been studied, either in the RRM domain of PPIE or the U-box motif of PPIL2, these domains contribute additional functionality to the isomerase they are encoded with. Although recent structures of splicing complexes show us intriguing glimpses into the roles of the WD40-domain or the RRM motif in mediating spliceophilins interactions, the proteomics data does not discriminate between domain-specific interactions. Further work on isolated, purified proteins will be needed to truly understand the roles of the additional isomerase domains on protein–protein interaction and/or function.

As an example of the limitations that all these concerns place upon the researcher studying protein:protein interactions with the spliceophilins, consider the case of PPIL2. Multiple proteins are predicted to bind to PPIL2 that we see binding to other spliceophilins in multiple proteomics studies. These include CRNKL1, along with ZNF830 and PRCC. By constructing a family of constructs to parse out domain and full-length protein:protein interactions, we have shown in vitro that PPIL2 interacts at a minimum with ZNF830, yielding both a quantitative validation of proteomics data, and an interesting and complex study of two multidomain proteins in solution (Figure 7). However, without visualization of PPIL2 and ZNF830 within the spliceosomal subcomplex they both participate in, we cannot say with certainty that the data being generated from these other experimental sources are indeed relevant to spliceosome biology. Similarly, proteomics-derived data for PPIL3 will not be described here, as it has not yet been validated by either in vitro approaches, nor visualized in spliceosome structure.

On the other hand, there are several cases presented in this review in which splicing complexes are providing the first independent validation of predicted interactions with spliceophilins. For PPIE, SF3B4 is indicated for the first time to be a proline-mediated interactor through the S1 pocket. Other proteins including SF3B2, SF3A2, snRNPA’, B/B’, and SmD3 all interact with the isomerase domain of PPIE. These structures are also able to isolate PPI interactions from those of RRM, with the RRM interacting with SF3A2, SYF1, AQR, and pre-mRNA substrate can be added to the list of PPIE interactions (Figure 5). All of these interactions are found within BioGRID, along with many others (CDC5L, SNW1, PRPF8, XAB2, etc.). As both the RRM and isomerase domains of PPIE are quite soluble and well-behaved in solution, perhaps the spliceosome structures will provide the impetus needed to drive further research into validation of these interactions individually in vitro. As this is the only spliceophilin seen interacting directly with pre-mRNA substrate, it is hoped that further work to identify the potential role of PPIE in regulating alternative or constitutive splicing in vivo will be inspired by this structure. PPIL1 is seen to interact with PRPF17 through S1. Interactors of PPIL1 include SNW1, CRNKL1, RBM22, CDC5L, U5 snRNP 40 kDa, and SYF2, vastly expanding the potential interaction repertoire of this minimal spliceophilin (Figure 6). Save for SNW1, most of these interactions are not contained within BioGRID or STRING, except for CDC5L. This highlights the importance of complementary approaches to obtain information about these protein:protein interactions within subcomplexes, such as the PRP19 complex. Interactions with U5 snRNP 200 kDa, BUD31, PRPF8, and RNF113 are seen in the CWC27-containing structures. It is interesting that CWC27, which is not an active isomerase like PPIE and PPIL1, is not seen interacting with any protein in the S1 pocket (although, as noted above, negative results cannot be conclusive due to the poor affinity proline-containing substrates have for both active isomerases, and even more so for substituted S1 sites) (Figure 9). None of these interactions were pulled out from STRING or BioGRID, at least not as direct interactions with CWC27. It is unfortunate that the low-complexity regions of CWC27 could not be modeled in structure, but many new hypotheses can be derived and tested from the current structural data, which is quite exciting. We find PPWD1 interacting with PRPF8 through both its PPI and WD40-domains, which also interacts with U5 snRNP 200 kDa (Figure 10). These interactions have not been captured in BioGRID or STRING, again generating new possibilities for functional and/or structural work. We note especially the power of visualizing the WD40-domain of PPWD1 in the spliceosome, as this isolated domain is historically extremely difficult to express and purify in vitro. Finally, PPIG interacts with SNW1, CWC15, and PRPF8, all through the isomerase domain (Figure 11). None of these interactions are in BioGRID or STING. The SR region of PPIG is not modeled, unfortunately; this region, which should be a hotspot for regulation of alternative splicing through SR-containing protein kinases, would have been extremely interesting to visualize, and is not likely to be ordered or properly post-translationally modified using in vitro purified protein.

Finally, two useful analyses to perform across all spliceosome structures are to look at the interactors seen to bind to spliceophilins over all structures. We see interactions through the S1 pocket by SF3B4 and PRPF17, and many interactions with the loop structures of the various spliceophilins. However, there are a few spliceosomal proteins that occur in multiple structures, interacting with multiple spliceophilins. Of note is the central splicing regulator PRPF8, which is in close proximity to four spliceophilin isomerase domains: PPWD1, CWC27, PPIG, and PPIH. This speaks indirectly to the central importance of the spliceophilins family in proper spliceosome function, and merits further research. Another interesting note is to compare the spliceophilins themselves. The structural similarity of the isomerase domain confounds researchers attempting to predict potential unique binding partners within the nucleus—across the eight nuclear cyclophilins, structural alignments result in an overall root mean squared deviation of less than 2 Å [1]. More importantly, it has become clear over the last 10 years in this field that the active site for prolyl isomerization is not the only, nor perhaps the major, site of protein–protein interactions with the spliceophilins [32,54]. One obvious outcome from the splicing complexes released to date is that we still do not understand how S1 interactions with proline-containing substrates are driven, at least in any way that could be predictive. There is no other corroborating evidence linking PRPF17 or SF3B4 as substrates for proline turnover, and yet the data from the spliceosome structures of mature B_act_ is quite convincing. Other than for PPIL2 or CWC27, in which the active site is substituted in such a way as to lose affinity for proline, we cannot see any differences between the S1 sites of spliceophilins that would preclude interactions with prolines, and there are thousands of proline-containing peptides in spliceosomal proteins [55]. It remains to be seen what the consequences of these interactions are, or under what conditions they can be studied in vitro. However, the mere presence of these interactions within B_act_ represent a wealth of information for cyclophilin researchers. Likewise, based on this collection of structures it seems that some spliceophilins are more promiscuous in their loop region interactions than others. The early spliceophilins PPIE and PPIL1 participate in a complicated and large protein network within precatalytic and catalytic complexes, while others (PPIG, PPIH, and PPWD1) are interacting with only one or two other proteins. As mentioned previously, one must be careful overanalyzing negative results in these structures, as disorder or dynamic heterogeneity may result in the inability to model protein or RNA into structure; however, from what we currently know, it seems that certain spliceophilins may be more connected in structure than others. What effect this may have on their biological function within splicing complexes is an exciting and largely unaddressed question which these structures may help to inspire.

## 4. Spliceophilin Function

It is important to consider what is known (or supposed) about the cellular role of the cyclophilins in biological function before considering rational drug design against this family of enzymes. With at least 18 family members in human cells, most ubiquitously expressed across tissues and throughout development and into adult, finding unambiguous biological effects due to a particular cyclophilin is difficult. The high degree of conservation in the active site of these proteins does not often allow for substrate proline prediction, and in the cases where biologically relevant substrates have been identified, it is often due to localization of a single or subset of cyclophilins away from the rest. For instance, in the identification of the secreted cyclophilins (PPIA, PPIB, PPIC) as interacting viral proteins, a critical determinant of specificity was the fact that these cyclophilins were the only ones that could reasonably be expected to be found outside of the cell. Likewise, the wealth of research identifying PPIF as a mediator of mitochondrial function, and therefore a viable drug target to modulate the same, benefitted from the fact that there is only one possible cyclophilin candidate in mitochondria. The identification of the nuclear cyclophilins as spliceophilins, and later assignment to cyclophilins to subcomplexes throughout the splice cycle, may well provide another opportunity to assign unique functions to individual cyclophilins. A spliceosome-wide approach using siRNA pools found that the spliceophilins, and especially the early spliceophilin PPIH, was actively participating in regulating the splicing of targets in apoptosis and inflammation [67,68]. We wished to see transcriptome-wide effects on splicing, and created stably transfected knockdown human cell lines, validated for the significant knockdown of each of the eight spliceophilins. We have completed splicing microarrays for each knockdown and can say that removing individual spliceophilins from cells results in large changes in thousands of splicing events (Figure 12A,B; see accession codes section for deposited Gene Expression Omnibus accession codes). We see, as in Papasaikas et al. [67] that when a subset of regulated splicing events are selected for analysis, there are certain spliceophilins that seem to exert greater regulatory control than others. Like Papasaikas et al. [67], we find that PPIH is a master regulator of alternative splicing; we also find that PPIL2 is in the same class, at least for the subset of splicing events and spliceophilins we studied (Figure 12C). By transiently transfecting expression constructs of isomerase domains into these stable lines, we can identify which splicing events are regulated more directly, over shorter timeframes; and by transfecting in isomerase domains with null mutations, we can assign events to isomerase activity (Figure 12B). Generally, what we see for the spliceophilins corroborates the models that we have generated from our in vitro studies; it is interactions outside the active site of the spliceophilins that largely dictate their ability to modulate splicing, not their isomerase activity per se (Figure 11B). On the other hand, we have seen several cases in which the isomerase activity of the spliceophilin is indeed dictating function; surprisingly, all effects to date have been on transcriptional targets. This was supported recently by an elegant study delineating the role of a subset of nuclear cyclophilins in directly interacting with the transcription factor brain and muscle aryl hydrocarbon receptor nuclear translocator-like protein 1 (BMAL1) and regulating circadian rhythm in cells [31]. We feel this is likely to be a more general mechanism; taken together, the effect of cyclophilins in the nucleus is to directly (through interaction with transcription factors) or indirectly (through chromatin) regulate transcription, and to indirectly modify alternative splicing through participation in splicing complexes mainly through interactions outside of the S1 pocket. Much work remains to be done on the biological function of spliceophilins within alternative splicing; this work will only benefit from the insights provided by spliceosomal structures and could also greatly benefit from an increased effort to design splicing modulators that target spliceophilins.

## 5. Spliceophilins: Highly Druggable Modulators of Nuclear Function?

It is worthwhile to discuss the state of the field regarding rational drug design to obtain isoform-selective cyclophilin inhibitors. For many years, the pan-inhibitor cyclosporine was the only available option for researchers to study cyclophilin activity in vitro or function in vivo; solubility issues and broad, potent inhibition of most cyclophilin family members presented many barriers to insight. The alternative scaffold of sanghliferin and later scaffolds have presented similar issues, often without an impressive increase in selectivity. Most drug development has focused on the connection between secreted cyclophilins and infectious disease, or the identification of PPIF as a modulator of the mitochondrial permeability transition pore complex [6,7,70]. The majority of this work continues to target the S1 pocket; highly conserved and extremely accessible to solvent, this approach is nearly guaranteed to generate pan-inhibitors with significant off-target activity [8,71,72]. In 2010, we published a study in which we identified and characterized a site contiguous to S1. This site, the S2 region, is much less well conserved across the cyclophilin family, and importantly there is often little or no conservation in S2 for cyclophilins localized to the same compartment of the cell [1]. A study published in 2016 was the first to use this analysis of S2 as a basis for the rational design of drug-like compounds, targeting PPIF [8]. The S2 approach seems primed to provide cyclophilin researchers with valuable pharmacophores to tease out the functional roles of individual cyclophilins in cells. The exponential growth of available spliceosomal complex structures, along with the incremental work helping to define spliceophilins as regulators of both splicing and transcription, presents a novel opportunity for research. Developing S2-selective small molecules to modulate alternative splicing, chromatin regulation, and transcriptional activity by spliceophilins could benefit both basic and translational research. If S2 targeting continues to be successful and is implemented in a systemic way to the spliceophilins subfamily, we may well be on the cusp of a new era of research into this complex, intriguing, and important class of human proteins.

## 6. Conclusions and Final Thoughts

The advent and proliferation of published cryo-EM structures of the spliceosome might have been expected to yield great structure-based insight into the spliceophilins. Indeed, a whole host of novel, hypothesis-generating observations can be made based on the models released to the public. It is interesting at this stage of spliceosomal research to review the unforeseen reasons these studies have not yet delivered completely on their great promise for those in the cyclophilin field. In all structures reported to date, a majority of the appropriate spliceophilins are retained during purification, as evidenced by mass spectroscopy; however, even when binding partners are partially visible, no clear density for the spliceophilin is seen. Although the cyclophilins are highly structured with very little conformational dynamics on long timescales, they participate in highly dynamic interactions within the spliceosome. Whether these conformational dynamics are imparted by cyclophilins interacting directly with intrinsically disordered proteins (in the cases of PPIL1 and SNW1, or PPIH and PRPF4); because the nature of PPI activity is to impart greater conformational heterogeneity in the region surrounding the target proline; or whether it is because spliceophilins tend to be found on the more dynamic periphery of the spliceosome, the effect is to make regions of the spliceosome containing cyclophilins more difficult to capture and visualize using structural methods. When these regions can be accurately modeled, they are providing great insight into the nature of the interactions between spliceophilins and their physiologically relevant binding partners; we await with great excitement future structures, which we expect will provide only more fruitful data for hypothesis generation in this field. Structural biologists have always led the way in studying the family of human cyclophilins and have consistently driven the conversation about cyclophilin function into new and interesting areas of research. Smaller studies on individual spliceophilins complexed with one or two binding partners remains the most tractable system for studying both orthosteric and allosteric interactions in higher resolution and on shorter timescales, but larger-scale structural work on spliceosomes and other nuclear complexes can also provide important insights. Working together with cell biologists to find the direct and indirect molecular targets of spliceophilin function in the nucleus, and chemists to design effective small molecules targeting the spliceophilins, will drive forward research into this intriguing, challenging, and rewarding field.

## 7. Accession Codes

Accession codes for the cyclophilins described in this manuscript (UniProt AC): PPIA (P62937), PPIH (O43447), PPIE (Q9UNP9), PPIL1 (Q9Y3C6), PPIL2 (Q13356), PPIL3 (Q9H2H8), CWC27 (Q6UX04), PPWD1 (Q96BP3), and PPIG (Q13427). The PDB ID codes for the structures presented in this review are highlighted throughout the text, and are also included here: PPIA (2CPL), PPIE (1ZMF, 2R99, 3UCH, 5MQF), PPIG (2GW2, 2WFI, 2WFJ), PPIH (2MZW), PPIL1 (1XWN, 2K7N, 2X7K), PPIL2 (1ZKC), PPIL3 (), CWC27 (2HQ6, 4R3E), PPWD1 (2A2N), B complex (5O9Z), early B_act_ complex (5Z58), B_act_ complex (6FF4), mature B_act_ complex (5Z56), late B_act_ complex (5Z57), C Complex (5YZG), C* 2^nd^ step complex (5MQF), C* complex (5XJC). GEO accession numbers for splicing microarray data resulting from knockdown of nuclear cyclophilins compared to scramble controls are: PPIH (GSE103648), PPIE (GSE117178), PPIL1 (GSE117381), PPIL2 (GSE117373), PPIL3 (GSE117302), CWC27 (GSE117144), PPWD1 (GSE117376), and PPIG (GSE117234).

## Figures and Tables

**Figure 1 biomolecules-08-00161-f001:**
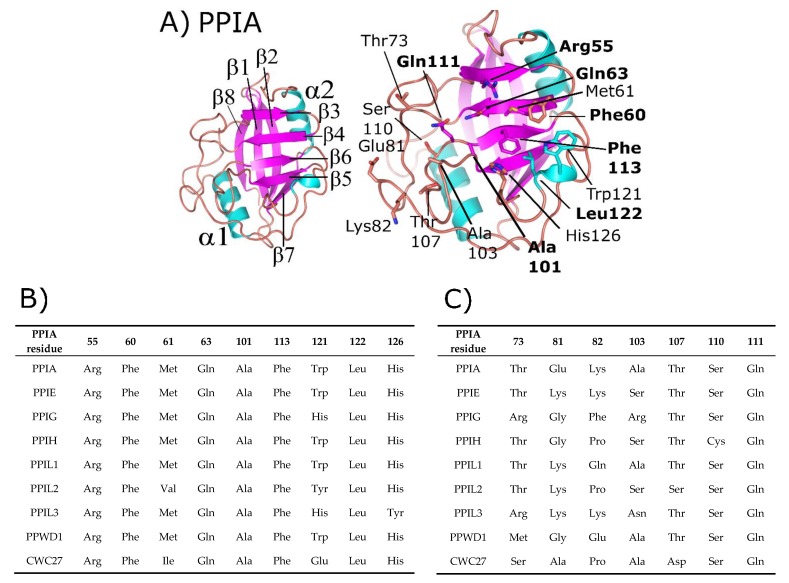
Structure and annotation of the canonical cyclophilin PPIA. (**A**) Left panel, secondary structural elements are labeled for the PDB ID: 2CPL. Right panel, residues that comprise the S1 (proline-binding) and S2 (specificity) pockets are shown in stick representation and labeled. S1 pocket labels are on the right, S2 pocket labels on the left, and in bold are residues invariant across the nuclear cyclophilin family. Residue numbering also follows PPIA, by convention. (**B**) The residues that comprise the S1 pockets are largely invariant, while (**C**) the residues that comprise the S2 pockets are more variable.

**Figure 2 biomolecules-08-00161-f002:**
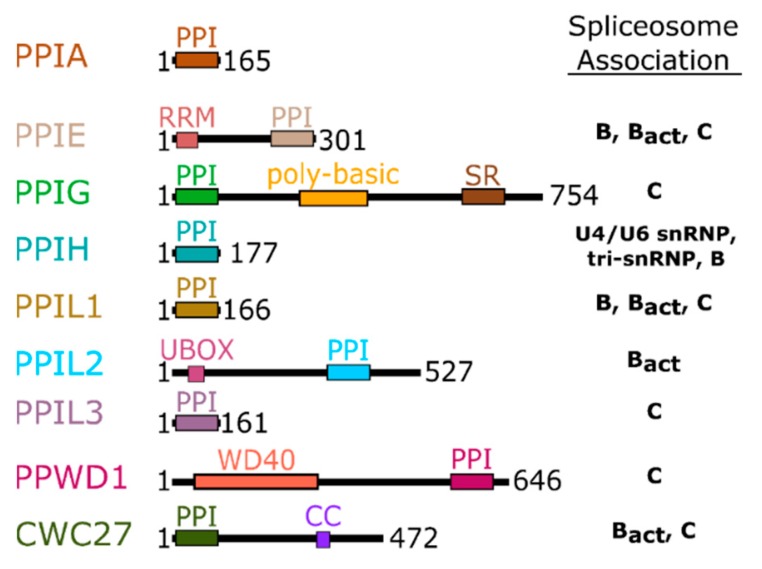
Domain organization of the nuclear cyclophilins. Colors are consistent throughout all figures. Spliceosome association is modified from [17,18] and is further shown in Figure 3.

**Figure 3 biomolecules-08-00161-f003:**
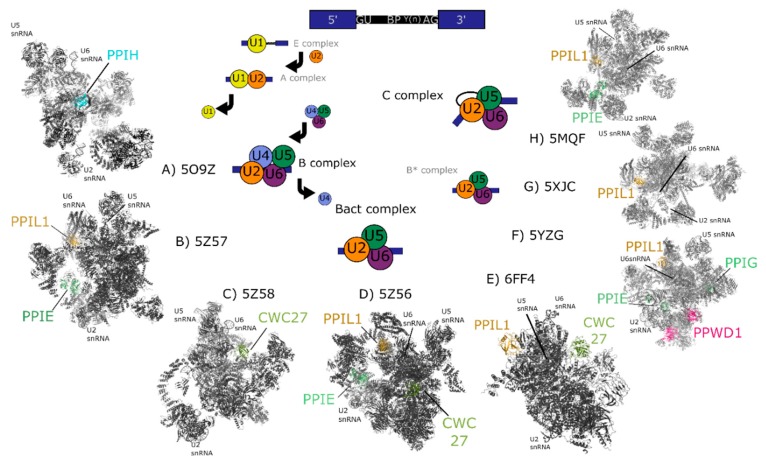
Cryo-electron microscopy structures of spliceosomal complexes. Center, a graphical representation of the canonical metazoan pre-mRNA substrate, including the 5′ and 3′ exons, the consensus 5′ and 3′ splice recognition sites, the presence of the branchpoint (BP) that forms the intermediate splicing lariat structure, and the polypyrimidine tract (Y(n)). The classical graphical representation of the splice cycle, shown in the center of the figure, follows the association and dissociations of uridine-rich small nuclear RNAs (U-snRNAs) to characterize the splicing intermediate (E, A, B, B_act_, B*) and catalytic (C) complexes. Complexes discussed in the text are shown on the outside of the splice cycle, with spliceophilins labeled and highlighted in color. The key U snRNAs U2, U5, U6 are present in all structures, and their rough positions labeled in each structure. The structures are roughly aligned to the part of the splice cycle they are thought to represent: B complex (PDB ID: 5O9Z) [24], B_act_ (PDB IDs: 5Z57, 5Z58, 5Z56, and 6FF4) [25,26], or C (PDB IDs: 5YZG, 5XJC, and 5MQF) [27,28,29].

**Figure 4 biomolecules-08-00161-f004:**
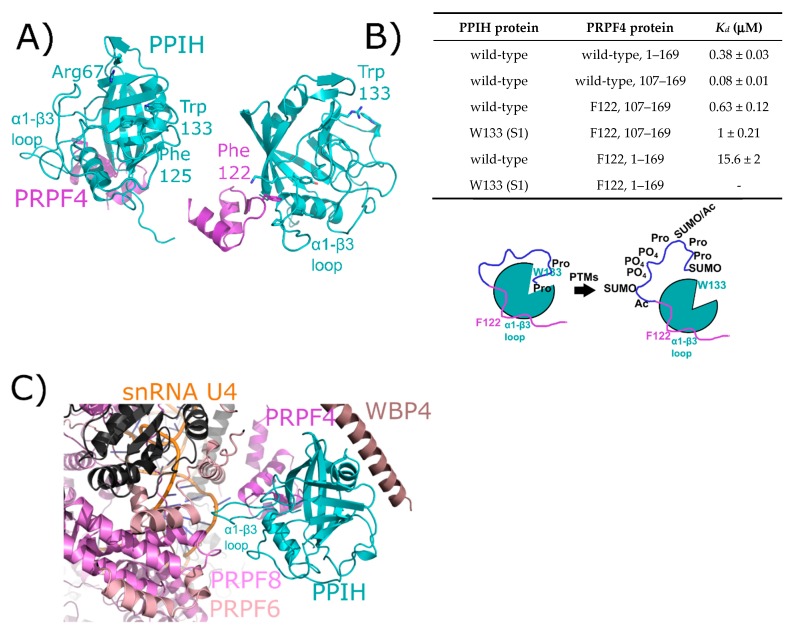
The interaction between PPIH and PRPF4 seen in solution is preserved in early spliceosomes. (**A**) The complex between a small peptide from PRPF4 and the PPIH isomerase domain is shown in cartoon representation (PDB ID: 2MZW) [40,41,42]. Key catalytic site residues are highlighted, and in the right panel the high-affinity interaction between Phe122 and the α1–β3 loop of PPIH is shown. In (**B**), a table summarizing the high- and low-affinity sites between PRPF4 and PPIH. Below, a model of the proposed function of the high- and low-affinity sites. Panels (**A**,**B**) are modified from their original form in [32]. (**C**) The neighborhood around PPIH in the B complex human spliceosome (PDB ID: 5O9Z). In addition to the previously known interaction with PRPF4, potential interactions with WBP4, PRPF6, and PRPF8 are highlighted. U4 snRNA is within 10 Å of PPIH and is labeled. Proteins in dark grey are further than 10 Å away from PPIH and are not candidates for direct interaction, so are not labeled. The region of PRPF4 forming the low-affinity interaction with PPIH (the extreme N-terminal 100 residues) is disordered in this structure. Ac: Acetylation; SUMO: Small Ubiquitin-Like Modifier.

**Figure 5 biomolecules-08-00161-f005:**
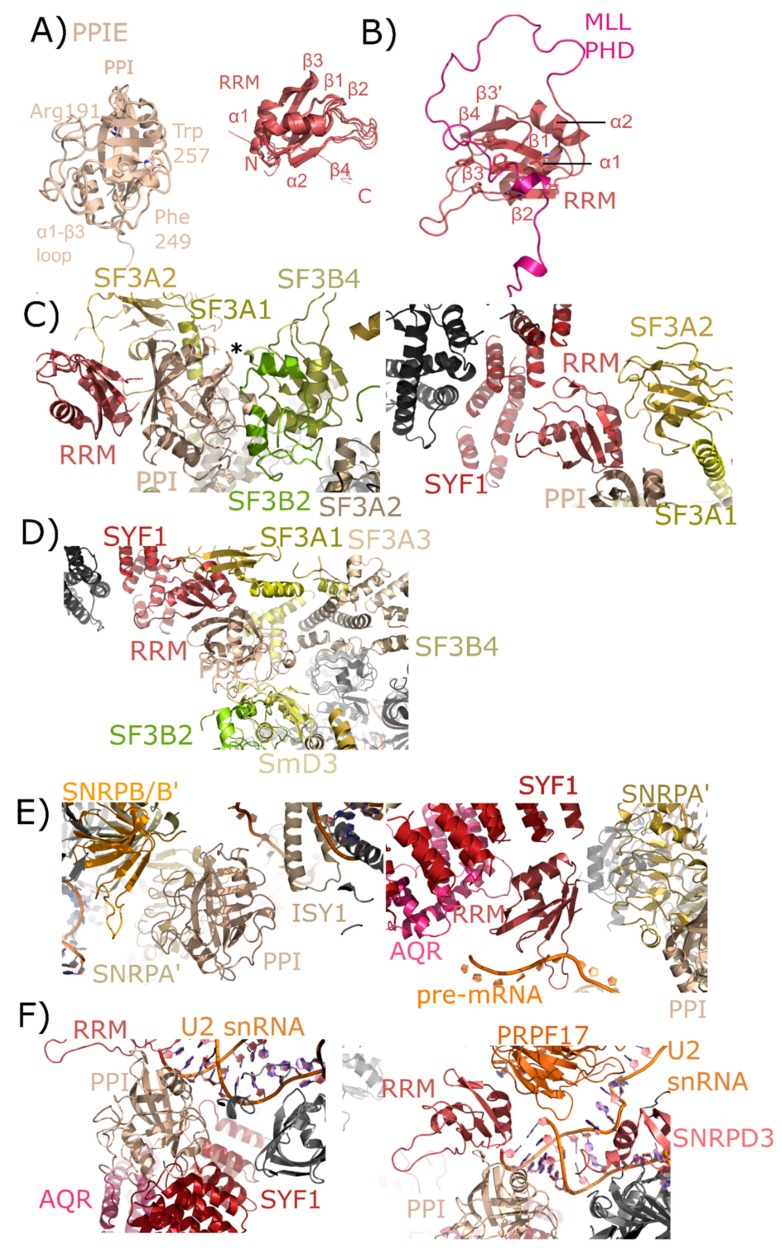
Structures of PPIE in solution and in spliceosomes. In (**A**), the cyclophilin and RRM domains of PPIE are shown in cartoon representation. The isomerase domain is represented by an overlay of PDB IDs: 2R99, 1ZMF, and 3UCH. Selected catalytic residues and protein–protein interaction regions are labeled. The RRM domain is represented by an overlay of PDB IDs: 2CQB, 2KU7, 2KYX, 3LPY, and 3MDF (chain A only). Secondary structure elements are labeled. (**B**) The RRM of PPIE was previously shown to interact with the PHD domain of MLL through an extensive interface involving multiple β-strands. (**C**) The neighborhood of the PPIE PPI (left) and RRM (right) domains in the mature B_act_ spliceosome is shown (PDB ID: 5Z56). The region around Pro83 of SF3B4 is pointed towards the active site of PPIE (marked with an asterisk). Other interactors are in color and labeled. (**D**) The neighborhood of the PPIE PPI (bottom center) and RRM (top center) domains in the late B_act_ spliceosome is shown (PDB ID: 5Z57). The protein network surrounding PPIE is unchanged relative to those shown in (**C**), but in this view SF3A3 is visible, which is just within 10 Å of PPIE isomerase domain. (**E**) The neighborhood of the PPIE PPI (left) and RRM (right) domains in the C complex spliceosome (PDB ID: 5YZG). Highlighted on the left panel are interactions between the isomerase domain and snRNPA’, B/B’, and SmD3 along with ISY1. Visible in this structure is the position of the pre-mRNA substrate, which is interacting with the RRM motif. In the right panel, other RRM interactors including SYF1, AQR, and snRNPA’. The isomerase domain is visible in the lower right corner for reference. (**F**) The neighborhood of the PPIE PPI (left) and RRM (right) domains in the C* complex spliceosome (PDB ID: 5MQF). In this model, the interaction with SYF1 and AQR is mediated through the isomerase domain of PPIE rather than the RRM motif, and the model of U2 snRNA is placed extremely close to PPIE. In this structure the orientation of PPIE seems to be flipped, with the isomerase domain interacting with AQR and SYF1, and the RRM motif close to PRPF17.

**Figure 6 biomolecules-08-00161-f006:**
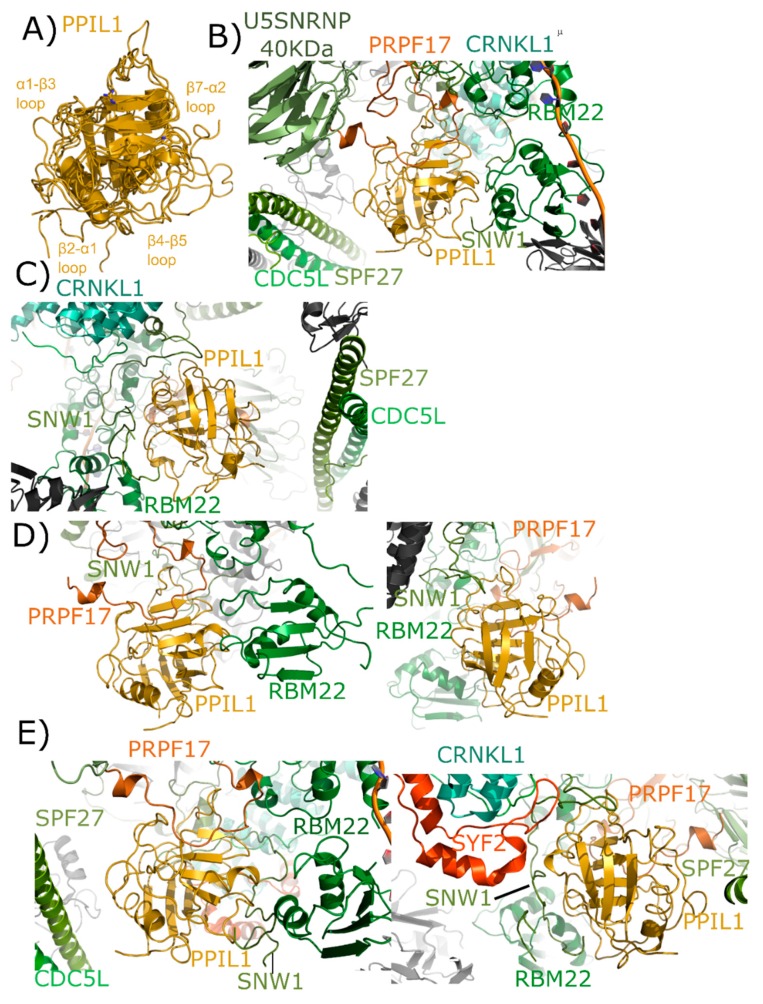
Structures of PPIL1 in solution and in spliceosomes. In (**A**) PPIL1 is shown in cartoon representation (overlay of PDB IDs: 1XWN, 2K7N, and 2X7K). Selected catalytic residues and protein–protein interaction regions are labeled, including the regions proposed to interact with SNW1/SKIP in solution. (**B**,**C**) Mature B_act_ complexes contain PPIL1 (PDB ID: 5Z56). (**B**) The view from the catalytic face of PPIL1. Proline 95 from PRPF17, which is centered in the S1 pocket, is highlighted. (**C**) The view from the back-face of PPIL1. The interactions between PPIL1 and SNW1, RBM22, CRNKL1, and CDC5L are more clearly seen. Note the extensive ordering of the disordered region of SNW1 around the β4–β5 and β7–α2 loop of PPIL1. The PPIL1 region in the late B_act_ complex (PDB ID: 5Z57) is identical to that of PDB ID: 5Z56, and is not shown here. (**D**) The PPIL1 neighborhood in the B_act_ complex (PDB ID: 6FF4). While many of the interactions are similar to those shown in (**B**,**C**), many of the proteins seen in that complex are missing from this model. Left panel, the view from the catalytic face; right panel, the view from the back face. Models of CRNKL1, CDC5L, and SPF27 are not included, and the ordered region of SNW1 is decreased. (**E**) The PPIL1 neighborhood in the C complex (PDB ID: 5YZG) centered on the catalytic face (left) and the back-face (right). The PPIL1 interaction neighborhood is similar to that shown in (**B**,**C**) with the addition of the SYF2 protein to the model, which interacts with the back-face of PPIL1 near α2. The models of C* complex in 5XJC and 5MQF are not shown, as the interaction environment around PPIL1 is very similar to that in (**E**).

**Figure 7 biomolecules-08-00161-f007:**
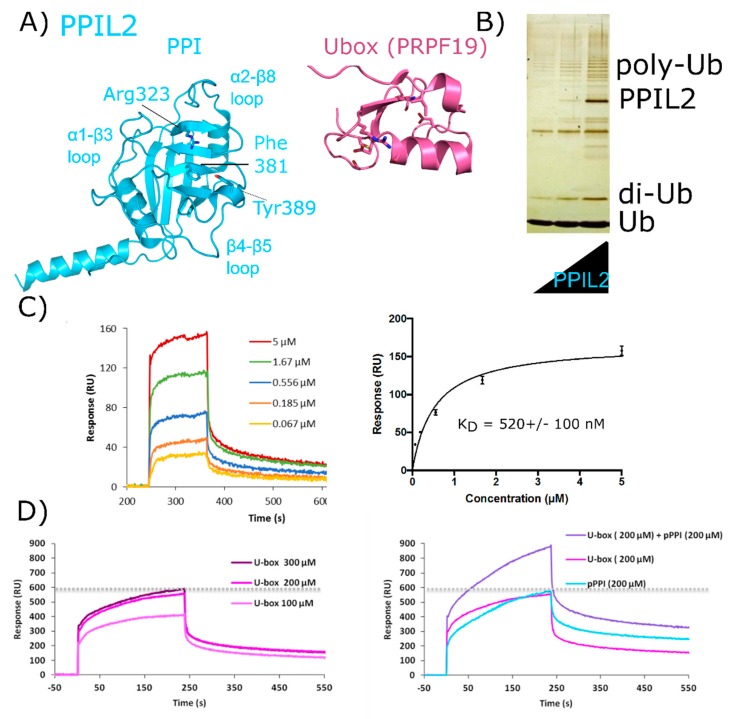
Validation of interactions between PPIL2 and the spliceosomal protein ZNF830. In (**A**) PPIL2 is shown in cartoon representation (PDB ID: 2ZKC). Selected catalytic residues and protein–protein interaction regions are labeled. (**B**) The U-box containing PPIL2 protein is a functional E3 ligase. Increasing concentrations of PPIL2 (0.1–10 µM) in an in vitro assay and in the presence of ubiquitin, E1, and E2 proteins lead to the production of poly-ubiquitin chains [54]. (**C**) Surface plasmon resonance of full-length PPIL2 (ligand) and ZNF830 (analyte) indicates nanomolar-affinity binding between the two proteins. (**D**) left panel, concentration dependence of isolated PPIL2 U-box binding to full-length ZNF830. Right panel, sequential addition of PPIL2 U-box followed by a mix of PPIL2 U-box and PPIL2 isomerase domain results in increasing response upon addition of ZNF830 zinc finger domain. This is interpreted as an ability for both domains of PPIL2 to interact simultaneously with ZNF830 ((**C**) and (**D**) are from [61]). Methods are summarized in Appendix B.

**Figure 8 biomolecules-08-00161-f008:**
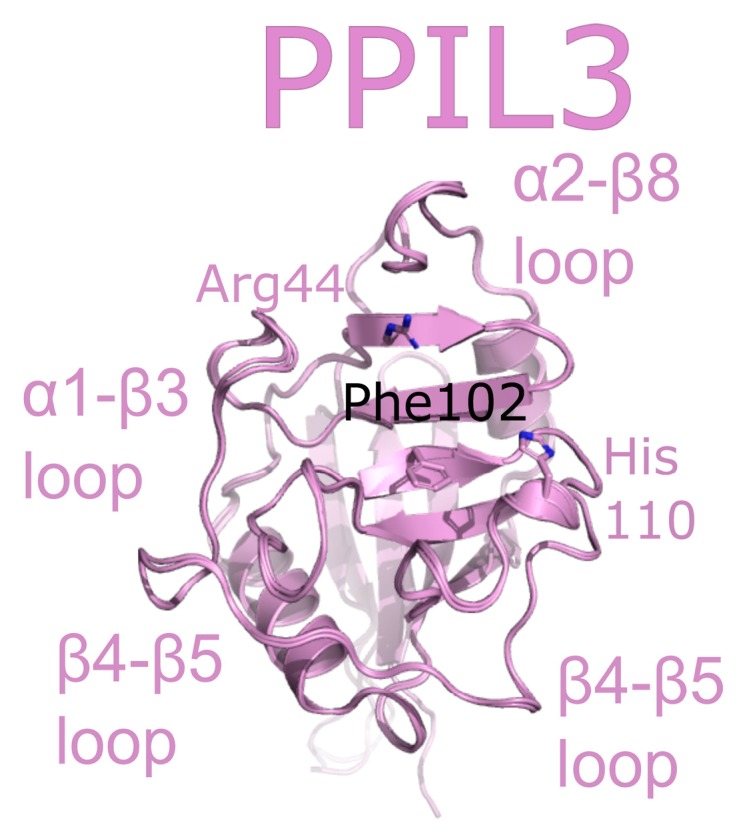
PPIL3 is shown in cartoon representation (PDB IDs: 2XWN and 2OK3). Selected catalytic residues and protein–protein interaction regions are labeled.

**Figure 9 biomolecules-08-00161-f009:**
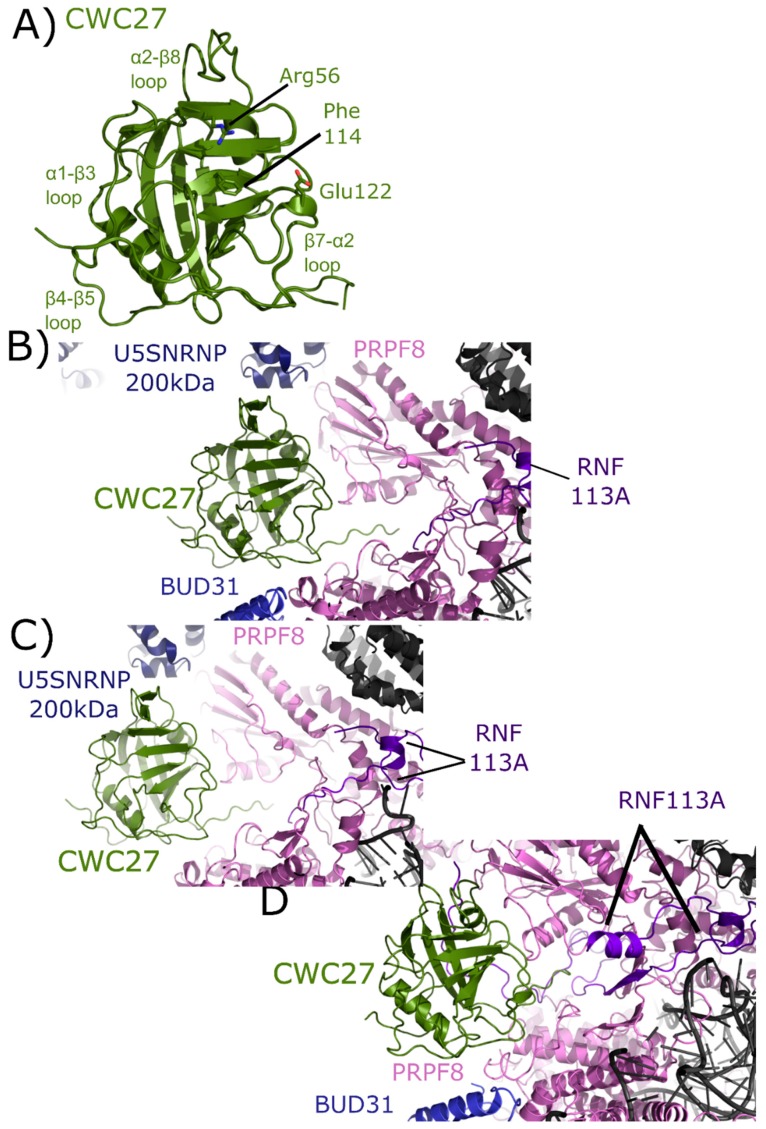
Structures of CWC27 in and out of the spliceosome. In (**A**) the isomerase domain of CWC27 is shown in cartoon representation (overlay of PDB IDs: 2HQ6 and 4R3E). Selected catalytic residues and protein–protein interaction regions are labeled. The substitution of Glu122 in the active site renders CWC27 inactive, although it still binds proline-containing peptides. In (**B**) the neighborhood around the isomerase domain of CWC27 in the mature B_act_ complex (PDB ID: 5Z56). Modeled interactions with U5 snRNP 200 kDa, BUD31, PRPF8, and RNF113 are highlighted. In (**C**), the neighborhood around CWC27 in the late B_act_ complex (PDB ID: 5Z58). The view is very similar to that in PDB ID: 5Z56, save for the absence of BUD31 and slight movement of PRPF8. In (**D**), the neighborhood around CWC27 in the B_act_ complex (PDB ID: 6FF4) is shown. Again, the modeled interactions are very similar to those in (**B**,**C**), with more of RNF113 modeled in 6FF4, including an additional, extensive interaction with β1–β2 of CWC27. The model includes BUD31 but U5 snRNP 200 kDa is missing. All models have only the isomerase domain of CWC27, with ≈200 additional residues uncharacterized.

**Figure 10 biomolecules-08-00161-f010:**
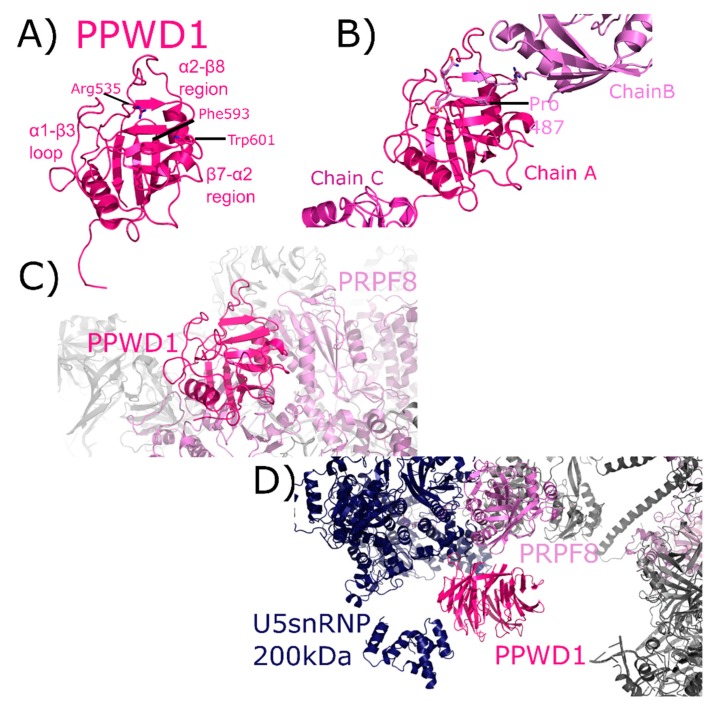
PPWD1 structures in and out of the spliceosome. In (**A**), one of the three molecules of the isomerase domain of PPWD1 from the asymmetric unit of PDB ID: 2A2N is shown in cartoon representation. Selected active site residues are labeled, along with the α1–β3 loop and the β7–α2 region. In (**B**), the view is expanded to include all three PPWD1 molecules in the asymmetric unit. The proline from a neighboring molecule is shown in the active site. Modified from [30]. In (**C**), the PPWD1 neighborhood is shown in the C complex (PDB ID: 5YZG). Only PRPF8 is close enough to form contacts with the isomerase domain of PPWD1. (**D**) Over 90 Å away, PRPF8 along with U5 snRNP 200 kDa are modeled near the WD40 domain of PPWD1. For reference, the region of PRPF8 that interacts with the isomerase domain of PPWD1 is visible in the upper-right corner of the figure, in color. This represents the first, and as of yet only, model of the WD40-domain of PPWD1.

**Figure 11 biomolecules-08-00161-f011:**
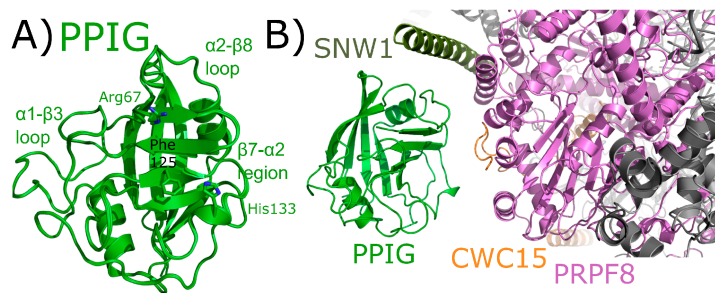
PPIG structures in and out of the spliceosomal context. In (**A**), the isomerase domain of PPIG is shown in cartoon representation (overlay of PDB IDs: 2GW2, 2WFI, and 2WFJ). Selected residues in the catalytic site, along with regions of protein–protein interactions, are labeled. In (**B**), the neighborhood around the isomerase domain of PPIG in the catalytic C spliceosome (PDB ID: 5YZG). Potential interactions between PPIG and SNW1, CWC15, and PRPF8 are highlighted. No model exists for the roughly 600 residues outside of the PPIG isomerase domain.

**Figure 12 biomolecules-08-00161-f012:**
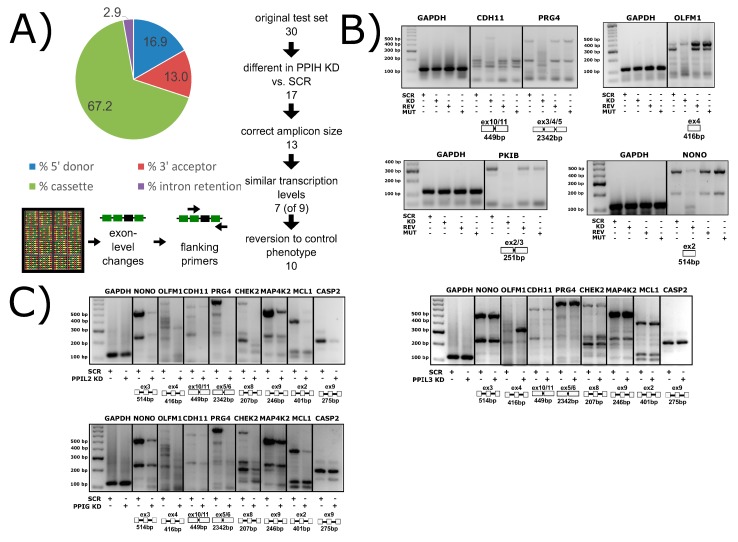
Spliceophilins impact alternative splicing patterns in human cells. (**A**) Knockdown of PPIH using RNAi results in changes in cassette exon, 5′ and 3′ splice sites, and intron retention, as measured by the Affymatrix HTA 2.0 array. (**B**) Compared to control cells (SCR, scrambled RNAi), PPIH knockdown cells (KD, PPIH-specific RNAi) exhibit altered alternative splicing of multiple genes (labeled at top of gels). Methods are summarized in Appendix C. When full-length PPIH is transiently expressed in knockdown cells (REV, PPIH ectopic expression), the splicing changes are reversed. When the W133A mutant of PPIH (MUT, PPIH mutant ectopic expression) is transiently expressed, the KD phenotype is seen, indicating that isomerase activity is dispensable for this function. (**C**) The knockdown of PPIL2 exhibits splicing patterns identical to that of PPIH knockdown, while PPIG only regulates a subset of genes, and PPIL3 does not regulate any of the common targets tested. Not shown, PPIH KD regulates all splicing events tested to date. All panels from [69]. Methods are summarized in Appendix C.

**Table 1 biomolecules-08-00161-t001:** Annotation of the nuclear cyclophilins.

Gene Name	UniProt Accession No.	# aa	Domain 1	Boundary	Domain 2	Boundary	Cryo-EM Structures
*PPIA*	P62937	165	PPI	1–165	-	-	-
*PPIE*	Q9UNP9	301	RRM	7–80	PPI	143–299	5MQF, 5YZG, 5Z56, 5Z57
*PPIG*	Q13427	754	PPI	11–176	SR/RS repeats	540–639	5YZG
*PPIH*	O43447	177	PPI	1–177	-	-	5O9Z
*PPIL1*	Q9Y3C6	166	PPI	1–166	-	-	5MQF, 5XJC, 5YZG, 5Z56, 5Z57, 6FF4
*PPIL2*	Q13356	527	U-BOX	42–101	PPI	281–433	-
*PPIL3*	Q9H2H8	161	PPI	1–161	-		-
*PPWD1*	Q96BP3	646	WD40	80–453	PPI	490–645	5YZG
*CWC27*	Q6UX04	472	PPI	14–166	coiled-coil	306–351	5Z56, 5Z58, 6FF4

# aa: Number of amino acids in the expressed, full-length protein. Domains are annotated as classified by SCOP and UniProtKB [37]. Domain boundaries are numbered according to amino acid residues.

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
