# Peer review of "Structural and Functional Insights into Human Nuclear Cyclophilins"

_biomolecules, 2018, doi:10.3390/biom8040161_

Round 1

Reviewer 1 Report

The manuscript by Rajiv et al., reviews our current state of knowledge on the structure and function of human nuclear cyclophilins and their roles in mRNA splicing. Cyclophilins are enzymes with peptidyl-prolyl isomerase (PPIase) activity and generally act as molecular chaperones aiding protein folding in vivo along with possessing a multitude of other functions. This review specifically focuses on the 8 major human nuclear cyclophilins involved in pre-mRNA splicing (termed spliceophilins) and discusses their structure and functions. Structures of these 8 nuclear cyclophilins, either alone or as part of the different spliceosomal complexes are analyzed using various crystal, NMR and cryo-EM structures. Interactions of each of the cyclophilins with spliceosomal proteins as observed in the cryo-EM structures are also discussed in detail. Furthermore, the manuscript talks about other functions of the nuclear cyclophilins and analyzes the wealth of structural information and large-scale proteomics data to provide insights into their interaction with other proteins and their important roles in splicing. Towards the end, the authors discuss the importance and challenges in developing specific small-molecule inhibitors for these proteins and assessing their functions, given their highly similar structures and functional redundancy. While similar reviews on the structure and functions of cyclophilins exist, the current manuscript contributes new knowledge by consolidating the structural and functional details specifically about the cyclophilins involved in splicing (spliceosomes) and by analyzing in detail the various recently solved crystal and cryo-EM structures. However, I think the manuscript in the current form is missing a few important details, the inclusion of which will make it accessible to a wider and more novice readers. For a structure based review, the organization and aesthetics of the figures in particular can be improved significantly. The review is certainly worth consideration for publication with some major modifications to the figures. Below are my detailed suggestions to improve the quality of the review

Major points:

1.     Given that the review discusses the roles of cyclophilins in the context of splicing and mentions the different spliceosome complexes multiple times throughout the text, it would benefit greatly from including a schematic figure of pre-mRNA splicing. This could be included as part of Fig. 2. Optionally, the involvement of spliceophilins at different steps in the splicing cycle could also be indicated in this figure.

2.     The specific suggestions on individual figures to make them more informative and improve their aesthetics are below.

3.     Figure 1: Color the structure of PPIA in A) by secondary structure. This will give a clearer view he different secondary structures and the overall fold of the cyclophilin family of proteins.

4.     Figure 1: A) Please use a contrasting color for the labels (such as yellow or a lighter color). Black on brown is not very visible. Also, it may be better to have the label on the side and point to the position of the residues with a line rather than writing directly on the structure which obscures it. Right panel: Color the residues of S1 and S2 using two different colors to clearly distinguish them spatially – please use contrasting colors as compared to the global structure, which can be shown in gray. Please point to the residues labeled with a black line. 

5.     Figure 1B,C: Showing a multiple sequence alignment of the proteins sequences belonging to a family with % conservation is a standard when comparing them. This will be more informative than simply listing the conserved residues and their positions in cylcophilins as a table. Optionally, the authors could also show a structural alignment of different PPIase structures (or of the eight spliceophilins discussed here) showing their conserved tertiary structure.

6.     Figure 2: As strongly suggested above, please provide a cartoon of the splicing cycle and then indicate the association of different cyclophilins with the spliceosome complexes. Since the review is in the context of their involvement in splicing, it will be greatly useful for the readers to get a brief overview of the pre-mRNA splicing cycle.

7.     Figure 3: A) Decrease the font size of the labels PPIH & PRPF in panel. Make of the same size as in panel C). B) Increase font size of text in the table and make the cartoon below bigger. In general, please follow a similar font size for labeling all the figures in the paper.

8.     Figure 4: Please use different colors for the various structures in the superposition in (A). Panels (A) and (B) need to be made bigger and as mentioned above, use black (or a contrasting color) font to label the residues or structural elements and point to their locations with black lines. Use similar font size as in panels C-F. In panels (B) and (E) – again use contrasting colors in a single panel. (B) magenta and brown are highly similar while in (E) red, magenta and brown look indistinguishable. Please use contrasting colors that are easy to visually distinguish in all the figures throughout.

9.     Figure 5: Panel (A) figure needs to be made larger and please label using a contrasting (suggestion, black) color. Please move the Phe113 to the side and point its position with a line.

10.  Figure 6: (A) PDB ID is incomplete or missing (B) Indicate concentrations of PPIL2 below the gel. (C) and (D) make the font size in the plots larger

11.  Figure 7: Could be combined with Figure 8, since this is a single panel figure

12.  Figure 8: A) Use different colors for the two structures overlaid- it was very hard to see that there are two structures in the figure. Cite 8A, D somewhere in the text – use different color for labels, use same font size B) pink and magenta are again indistinguishable.

13.  Figure 10: A) Please use different colors for the overlay – this will make the subtle differences visible.

14.  Figure 11: Could be made into a large full page image to be more clear.

Minor points:

15.  Comma is missing in the affiliations superscript between 1,2 and 1,3 in the title

16.  In the abstract and elsewhere in the text, the authors could refer to RCSB as RCSB PDB or simply PDB as more commonly used.

17.  Line 30: ‘evolutionally’ – typo, please correct it to evolutionarily

18.  Line 35: Sentence is redundant…This is the case in humans…

19.  Line 38: ‘PPIB and ….in various reports’ – please cite the relevant references here.

20.  Line 40: Add viruses at the end: hepatitis A, B and C viruses

21.  Line 86: and the title for Table 1: ‘Bioinformatics’ – not sure what Bioinformatics refers to here. Please change the label to ‘Selected details about the different on the nuclear cyclophilins’

22.  Lines 142-143: Initial studies referred to PPIE as Cyp33.

23.  Lines 145-147: Please indicate the resolutions for the structures, when describing them as modest and slightly lower resolution.

24.  Line 164: PPIE ‘is’ (missing) an early spliceophilin;

25.  Line 167: Rephrase the sentence as ‘Versions of all three of these complexes from S. pombe and human have been published.

26.  Line 370: Modified from [] – please insert the missing citation.

27.  Lines 538-540; ‘specific’ word is used 4 times in the sentence. Split into two simple sentences.

28.  In general, please refer to all the figure panels in the text. For example, panels 8A and 8D are not cited in the text. There are multiple examples of panels in others figures that are not referenced in the text.

Author Response

Response to Reviewer #1:

We thank the reviewer for their careful and very helpful review. We agree strongly with the majority of the suggested revisions and have attempted to address all of the suggestions by the reviewer. We have included details addressed to each of the reviewer’s points below.

Major points:

1.     Given that the review discusses the roles of cyclophilins in the context of splicing and mentions the different spliceosome complexes multiple times throughout the text, it would benefit greatly from including a schematic figure of pre-mRNA splicing. This could be included as part of Fig. 2. Optionally, the involvement of spliceophilins at different steps in the splicing cycle could also be indicated in this figure.

Indeed, this is a glaring omission on our part. We’ve decided to add a separate figure that both outlines the basics steps of a splicing cycle and includes global images of each of the spliceosomal structures outlined throughout the text with the position of the spliceophilins indicated. This is the new Figure 3. We have also given a more reasonable introduction to pre-mRNA splicing and the roles of the various spliceosomal complexes, found in the end of Section 1 (lines 74-112 in the final manuscript).

2.     The specific suggestions on individual figures to make them more informative and improve their aesthetics are below.

We addressed most of these suggestions in the Figures, whenever we could. See below for details.

3.     Figure 1: Color the structure of PPIA in A) by secondary structure. This will give a clearer view he different secondary structures and the overall fold of the cyclophilin family of proteins.

4.     Figure 1: A) Please use a contrasting color for the labels (such as yellow or a lighter color). Black on brown is not very visible. Also, it may be better to have the label on the side and point to the position of the residues with a line rather than writing directly on the structure which obscures it. Right panel: Color the residues of S1 and S2 using two different colors to clearly distinguish them spatially – please use contrasting colors as compared to the global structure, which can be shown in gray. Please point to the residues labeled with a black line.

Yes, we have adapted the Figure to address the issues outlined.

5.     Figure 1B,C: Showing a multiple sequence alignment of the proteins sequences belonging to a family with % conservation is a standard when comparing them. This will be more informative than simply listing the conserved residues and their positions in cylcophilins as a table. Optionally, the authors could also show a structural alignment of different PPIase structures (or of the eight spliceophilins discussed here) showing their conserved tertiary structure.

The high level of sequence and structure conservation in the cyclophilin family is a pretty well-traveled path, so we didn’t feel we had anything novel to add to that discussion. However, for completeness we did include a simple alignment across the cyclophilin domain, along with a structural alignment of the eight spliceophilins compared to the canonical cyclophilin PPIA. This is now an additional supplemental file (Supplemental Figure 1) and is referenced in the manuscript (lines 49-51).

6.     Figure 2: As strongly suggested above, please provide a cartoon of the splicing cycle and then indicate the association of different cyclophilins with the spliceosome complexes. Since the review is in the context of their involvement in splicing, it will be greatly useful for the readers to get a brief overview of the pre-mRNA splicing cycle.

See above.

7.     Figure 3: A) Decrease the font size of the labels PPIH & PRPF in panel. Make of the same size as in panel C). B) Increase font size of text in the table and make the cartoon below bigger. In general, please follow a similar font size for labeling all the figures in the paper.

Done.

8.     Figure 4: Please use different colors for the various structures in the superposition in (A). Panels (A) and (B) need to be made bigger and as mentioned above, use black (or a contrasting color) font to label the residues or structural elements and point to their locations with black lines. Use similar font size as in panels C-F. In panels (B) and (E) – again use contrasting colors in a single panel. (B) magenta and brown are highly similar while in (E) red, magenta and brown look indistinguishable. Please use contrasting colors that are easy to visually distinguish in all the figures throughout.

We have made most of the suggested changes. We respectfully disagree with the point made in (A) that the superposition panels should be in different colors. We feel the use of the same color makes our point more obvious; that the repeated structural determinations of these proteins result in virtually identical structure. And throughout the rest of the Figures, we are under time constraints that do not allow us to regenerate all the panels of all the figures in the manuscript. We believe keeping the color scheme consistent between structures, etc., outweighs any potential confusion for the reader. The figures are meant to point the interested reader back to the original structures and/or manuscripts referenced in the Review. Hopefully they reach that goal. We will note that in several cases, the colors are similar but there are other visual cues (i.e. RNA vs protein, etc.).

9.     Figure 5: Panel (A) figure needs to be made larger and please label using a contrasting (suggestion, black) color. Please move the Phe113 to the side and point its position with a line.

We have made changes to try and make the Figure more easily comprehensible.

10.  Figure 6: (A) PDB ID is incomplete or missing (B) Indicate concentrations of PPIL2 below the gel. (C) and (D) make the font size in the plots larger

We have made changes as suggested. (A) and (B) were addressed in the caption.

11.  Figure 7: Could be combined with Figure 8, since this is a single panel figure

To make Figure 8 a bit bigger, we will keep Figure 7 separate.

12.  Figure 8: A) Use different colors for the two structures overlaid- it was very hard to see that there are two structures in the figure. Cite 8A, D somewhere in the text – use different color for labels, use same font size B) pink and magenta are again indistinguishable.

See above. The figure has been modified to make the fonts more consistent. We will check the references to the Figures throughout the manuscript.

13.  Figure 10: A) Please use different colors for the overlay – this will make the subtle differences visible.

See above.

14.  Figure 11: Could be made into a large full page image to be more clear.

Yes, we agree.

Minor points:

We have addressed all the minor points listed below in the final version of the text and thank Reviewer #1 for their careful editing of the manuscript. We believe these suggested changes increase the clarity and potential impact of the manuscript and thank Reviewer #1 for their time.

15.  Comma is missing in the affiliations superscript between 1,2 and 1,3 in the title ok

16.  In the abstract and elsewhere in the text, the authors could refer to RCSB as RCSB PDB or simply PDB as more commonly used. ok

17.  Line 30: ‘evolutionally’ – typo, please correct it to evolutionarily ok

18.  Line 35: Sentence is redundant…This is the case in humans… ok

19.  Line 38: ‘PPIB and ….in various reports’ – please cite the relevant references here. ok

20.  Line 40: Add viruses at the end: hepatitis A, B and C viruses ok

21.  Line 86: and the title for Table 1: ‘Bioinformatics’ – not sure what Bioinformatics refers to here. Please change the label to ‘Selected details about the different on the nuclear cyclophilins’ ok

22.  Lines 142-143: Initial studies referred to PPIE as Cyp33. ok

23.  Lines 145-147: Please indicate the resolutions for the structures, when describing them as modest and slightly lower resolution. ok

24.  Line 164: PPIE ‘is’ (missing) an early spliceophilin; ok

25.  Line 167: Rephrase the sentence as ‘Versions of all three of these complexes from S. pombe and human have been published. ok

26.  Line 370: Modified from [] – please insert the missing citation. ok

27.  Lines 538-540; ‘specific’ word is used 4 times in the sentence. Split into two simple sentences. ok

28.  In general, please refer to all the figure panels in the text. For example, panels 8A and 8D are not cited in the text. There are multiple examples of panels in others figures that are not referenced in the text ok

Reviewer 2 Report

The review by Rajiv and Davis about human nuclear cyclophilins contains wealth of information about structures of cyclophilins as well as their abilities to interact with other proteins and oligonucleotides within the spliceosome. The authors have chosen eight human spliceosome cyclophilins and described them in relation to their functioning in the spliceosomal complex. The authors have gathered all or at least most of the accessible references and put them to the review.  The manuscript is well written without factual and formal errors.

According to my opinion, the manuscript can be published as it is. I have only one objection. Some of the figures are cluttered. It would be more illustrative to simplify them and make them larger, some of the details described in the text are often very difficult to follow on the figures.

Author Response

Response to Reviewer #2:

We thank the reviewer for their kind review. We agree that in some cases, the figures are a bit cluttered; and that the multi-panel figures (Figures 4 and 5) would be easier to interpret if larger. We have made these suggested changes and thank Reviewer #2 for their time.